# Anthropogenic ecosystem disturbance and the recovery debt

David Moreno-Mateos[1,2,3], Edward B. Barbier[4], Peter C. Jones[5], Holly P. Jones[5,6], James Aronson[7,8], José A. López-López[9], Michelle L. McCrackin[10], Paula Meli[3,11], Daniel Montoya[12,13] & José M. Rey Benayas[3,14]

Ecosystem recovery from anthropogenic disturbances, either without human intervention or assisted by ecological restoration, is increasingly occurring worldwide. As ecosystems progress through recovery, it is important to estimate any resulting deficit in biodiversity and functions. Here we use data from 3,035 sampling plots worldwide, to quantify the interim reduction of biodiversity and functions occurring during the recovery process (that is, the 'recovery debt'). Compared with reference levels, recovering ecosystems run annual deficits of 46–51% for organism abundance, 27–33% for species diversity, 32–42% for carbon cycling and 31–41% for nitrogen cycling. Our results are consistent across biomes but not across degrading factors. Our results suggest that recovering and restored ecosystems have less abundance, diversity and cycling of carbon and nitrogen than 'undisturbed' ecosystems, and that even if complete recovery is reached, an interim recovery debt will accumulate. Under such circumstances, increasing the quantity of less-functional ecosystems through ecological restoration and offsetting are inadequate alternatives to ecosystem protection.

[1] Basque Center for Climate Change–BC3, Edif. Sede 1, 1º, Parque Tecnológico UPV, Barrio Sarriena s/n, 48940 Leioa, Spain. [2] IKERBASQUE, Basque Foundation for Science, 48008 Bilbao, Spain. [3] Fundación Internacional para la Restauración de Ecosistemas, 28008 Madrid, Spain. [4] Department of Economics and Finance, University of Wyoming, Laramie, Wyoming 82071, USA. [5] Department of Biological Sciences, Northern Illinois University, DeKalb, Illinois 60115, USA. [6] Institute for the Study of the Environment, Sustainability and Energy, Northern Illinois University, DeKalb, Illinois 60115, USA. [7] Missouri Botanical Garden, St. Louis, Missouri 63110, USA. [8] Centre d'Ecologie Fonctionnelle et Evolutive (UMR 5175, Campus CNRS), 34293 Montpellier, France. [9] School of Social and Community Medicine, University of Bristol, Bristol BS8 2PS, UK. [10] Baltic Sea Centre, Stockholm University, 106 91 Stockholm, Sweden. [11] Natura y Ecosistemas Mexicanos A.C., Mexico DF 01000, Mexico. [12] Centre for Biodiversity Theory and Modeling, Station D'Ecologie Experimentale du CNRS, 09200 Moulis, France. [13] Centre INRA de Dijon, 21000 Dijon, France. [14] Departamento de Ciencias de la Vida, Universidad de Alcalá, 28871 Alcalá de Henares, Spain. Correspondence and requests for materials should be addressed to D.M.M. (email: david.moreno@bc3research.org).

Few ecosystems on Earth are undisturbed by people[1] and many degraded ecosystems are in the process of recovering worldwide[2–4]. Although in most cases the recovery process is without human intervention, societies spend billions of dollars annually to restore ecosystems[5–7]. Supporting recovery without intervention and repairing disturbed ecosystems are crucial to regain lost biodiversity, ecosystem functions and services provided to society[8–10]. Assessments of anthropogenic disturbances have shown global losses[11] in biodiversity, whereas the disturbance is still active and time lags exist in its response[12,13] (Fig. 1). However, as ecosystems recover after the disturbance ceases, it is less clear to what extent they continue to endure deficits in biodiversity and functionality.

Here we quantify the interim reduction of biodiversity and biogeochemical functions occurring during ecosystem recovery, which we call the 'recovery debt'. This metric measures the per annum amount that an ecosystem function or biodiversity is reduced during the recovery process after disturbance ceases (Fig. 1). The recovery debt is a useful indicator of the magnitude of ecosystem degradation, because even if ecosystems eventually recover their biodiversity and functions, there may be a long period of time until complete recovery is achieved. During the recovery debt period, shortfalls in biodiversity and ecosystem functionality will affect the quantity and quality of ecosystem services provided by the recovering systems.

## Results

**Meta-analysis descriptors**. We found data from 3,035 sampling plots from 348 published primary studies covering a total study area $>550,000\,\text{km}^2$ (Supplementary Figs 1 and 2, Supplementary References and Supplementary Table 1). Data collection was restricted to six major ecosystem categories (forests, grasslands, wetlands, rivers, lakes and marine ecosystems), eight anthropogenic disturbance categories (agricultural transformation, logging, mining, invasive species, eutrophication, hydrological disruption, overfishing and oil spills or combinations of them) and four recovery metrics (organism abundance, species richness, carbon cycling and nitrogen cycling). We also included hurricanes as an example of a natural disturbance for reference.

The outcome measures in the database related to the recovery metric 'organism abundance' included measurements of density,

biomass, cover and basal area of trees, shrubs, grasses and algae, and measurements of density of birds, fish and invertebrates. The outcome measures related to the recovery metric 'diversity' included mainly measurements of species richness and diversity indexes, such as Shannon, Simpson and evenness indexes. Biogeochemical outcome measures related to the cycling of carbon and nitrogen contain both pools and fluxes of these elements in soil, litter and the water column. We amassed 3,816 outcome measures for which two measures of recovery were collected over time and compared with a reference value. The reference value was taken from either the same ecosystem before degradation occurred or a nearby comparable ecosystem that was undisturbed.

**Recovery debt estimations**. A per annum recovery debt was found in all the categories in which data were available (Fig. 2). We found that ecosystems undergoing recovery had about half of abundance (46–51%, 95% confidence intervals of the mean effect size) and one-third of species diversity (27–33%) compared with reference values (Fig. 2a), over 22 and 16 years (average time since recovery started), respectively, following a disturbance. This pattern was markedly consistent across ecosystem categories, which did not show strong moderating effects on our models except for the abundance debt (Supplementary Table 2). However, we did find strong moderating effects in the disturbance categories studied (Fig. 2b). These results were not affected by the organism type (Supplementary Figs 3 and 4).

Carbon and nitrogen debts (32–42% and 31–41%, respectively) did not differ after 24 and 14 years of recovery, respectively. Ecosystems affected by eutrophication showed the highest organism abundance debts (52–63%; Fig. 2b) and nitrogen debts (35–51%) after 29 and 6 years, respectively. Formerly mined sites showed the highest diversity (32–45%) and carbon (39–62%) debts after 11 years (Fig. 2d). In ecosystems recovering from hurricanes, we found the lowest diversity, carbon and nitrogen debts after only two to seven years of recovery.

## Discussion

The consistent decrease in diversity and abundance found in recovering ecosystems may, at first glance, contrast with other studies showing that α-diversity does not change through time[14,15]. However, our recovery metric 'diversity' includes other diversity measurements that account for differences in abundance, which could be responsible for this contrast. Nonetheless, our results agree with the worst scenarios estimated for the effects of land-use change on local species richness of plants and animals[11], and with the reanalysis of references[14,15], showing that spatial and temporal biases in these meta-analysis do not support a no net change of α-diversity[16]. This highlights that species assemblages could be more resilient to anthropogenic disturbance than populations, even when most individuals are lost.

Although nitrogen recovery debts could be expected to be lower than carbon debts because of faster turnover rates of nitrogen[10,17], our results suggest similar impacts of anthropogenic disturbances in the cycling of both elements. This adds evidence to other large-scale recovery estimations that found similar recovery patterns for the cycling of carbon and nitrogen[18,19]. Our results also suggests that mining and water pollution, caused by agriculture and urban uses, could be not only major drivers of biodiversity, and ecosystem function and service loss[20,21], but also major drivers preventing their recovery. The fact that hurricanes were responsible for the lowest recovery debts suggests that the negative effects of anthropogenic disturbances could cause more pervasive damage than some natural disturbances.

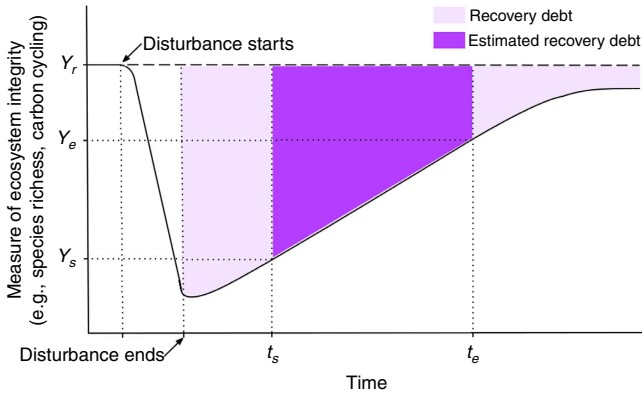

**Figure 1 | Measurement of the recovery debt.** The light shading represents the total amount that an indicator of ecosystem integrity (outcome measure, for example, biodiversity or an ecosystem function) is reduced during recovery after a disturbance ceases, that is, the recovery debt. The dark shading represents our estimation of the recovery debt between the time when the measurement of the outcome measure started ($Y_s$, $t_s$) and when the measurement ended ($Y_e$, $t_e$). The dashed line ($Y_r$) represents the reference goal value existing in the pre-disturbance state or in another ecosystem with similar conditions that remained 'undisturbed'.

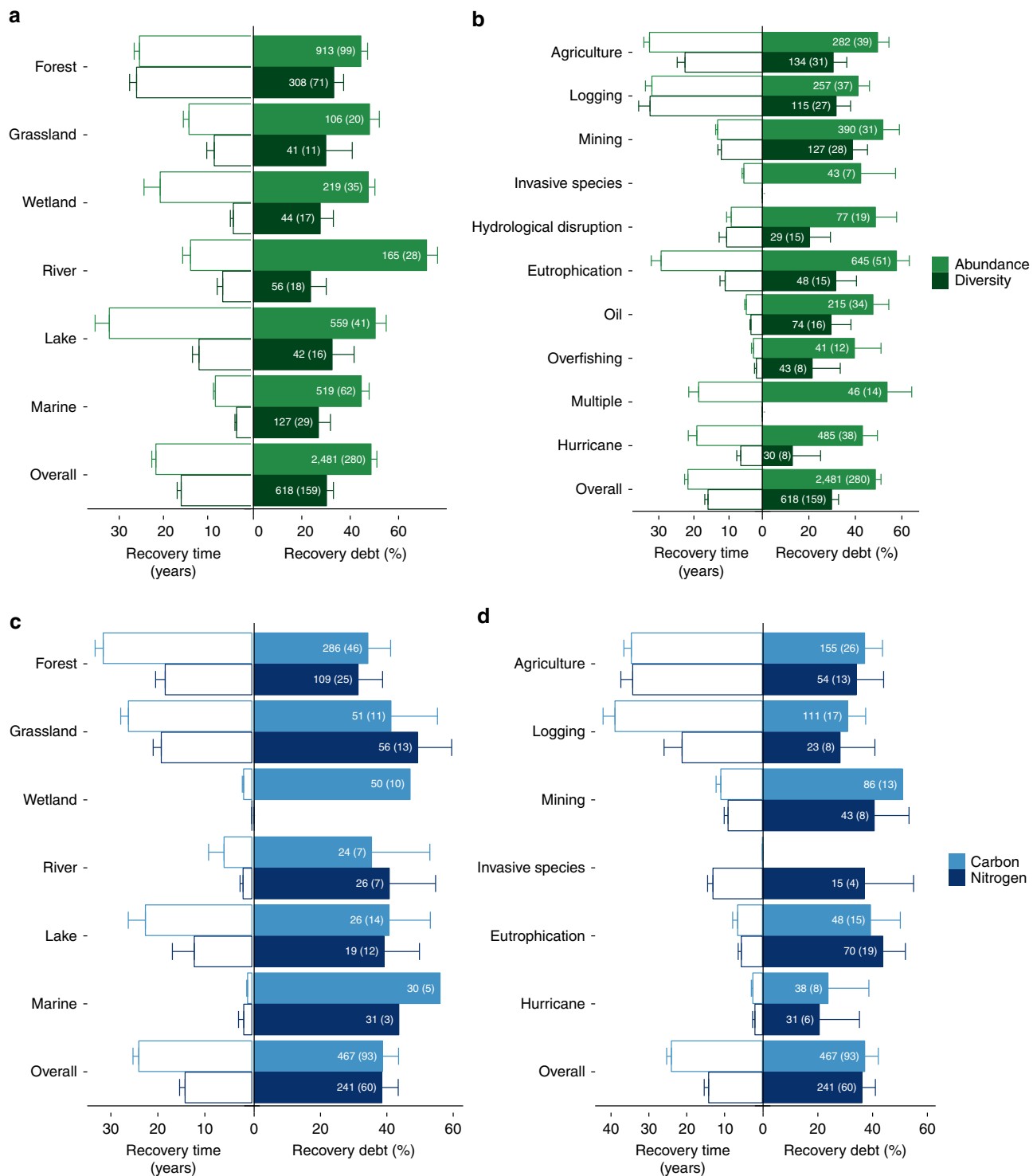

**Figure 2 | Recovery debt per annum estimated across ecosystem and disturbance categories.** Predicted means and 95% confidence intervals estimated by the generalized linear mixed models for the abundance of organisms, species diversity and cycling of carbon and nitrogen across ecosystem (**a**,**b**) and disturbance (**c**,**d**) categories. Recovery times on the left of each panel are the mean and s.e. of the time since recovery started associated to each recovery debt value. Numbers in bars are the numbers of outcome measures (and of studies).

Caution must be exercised on the interpretation of these results for three reasons: first, our results are based on a limited number of outcome measures of biodiversity and biogeochemical functions (Supplementary Table 1) selected to reduce the heterogeneity of data commonly associated with large meta-analyses. Consequently, our selected metrics of recovery are likely to be underrepresenting the complexity of ecosystems and thus

are conservative estimates of the complete magnitude of the recovery debt. Second, we detected substantial between-study heterogeneity in the meta-analysis, which could involve higher variance than the one we included in our models. Third, we have combined heterogeneous outcome measures including indicators of diversity of different life forms or different measures of carbon cycling and storage. Although these combined indicators may be

sometimes difficult to interpret, constructing such recovery metrics allows us to provide a consistent indication of the magnitude of ecosystem degradation from disturbance followed by recovery across a wide range of global ecosystems and anthropogenic disturbances.

Even under the assumption that ecosystems will eventually recover to their 'reference' values at longer time scales than are included in this study, our results reveal a consistent pattern: the interim per annum debt of abundance, diversity, and carbon and nitrogen cycling of degraded ecosystems across the globe is pervasive and continues for decades or more. Our findings support studies showing that complete recovery may not be achievable during decades or more[10,22,23] and similar outcomes might occur globally across multiple disturbances. Beyond previous estimates of the effects of disturbance on biodiversity loss and its time lags[11,13], these findings show that during recovery ecosystems worldwide have less plants and animals, and lower biodiversity and functions compared with undisturbed systems. In particular, recovering ecosystems may not only have lower diverse than undisturbed ones, but also may be much less populated.

These results suggests caution in pursuing ecosystem management strategies that exclusively rely on restoration or recovery to reverse biodiversity and functional loss[24–27]. This is particularly relevant in biodiversity offsetting strategies that allow ecosystem degradation if compensated through eventual restoration[28–30]. Given the lack of complete recovery, any further degradation, even if compensated by restoration, would increase the overall recovery debt of ecosystems. This would also suggest reconsidering restoration policies that attempt to fulfill 'no net loss' principles by simply increasing the mitigating or offset ratio so that more area of less-functional ecosystems are created[30]. If the restoration debt is large and sustained over several decades, then increasing the quantity of less-functional ecosystems is poor compensation for the overall intervening loss in ecosystem biodiversity and functions. Under such circumstances, ecological restoration and offsetting are inadequate alternatives to ecosystem protection.

## Methods

**Database construction.** The database is the result of merging two previously published meta-analytical databases[8,31] with a new and extended database. To create the new database, on May 2013, we did a simultaneous search in Web of Science and Google Scholar using the search chain '(agriculture OR damming OR eutrophication OR hurricane OR invasive species OR logging OR mining OR multiple OR 'oil spill' OR overfishing) AND (recovery OR resilience)' restricted to the research areas 'Agriculture, Biochemistry molecular biology, Environmental sciences ecology, Physiology, Toxicology, Biodiversity conservation, Developmental biology, Reproductive biology, Plant sciences, Geology, Fisheries, Forestry, Water resources, Marine freshwater biology, Microbiology, Parasitology, Entomology, Behavioral sciences, Geography, Zoology, Cell biology, Mycology, Paleontology, Archaeology, Demography, Physical geography, Evolutionary biology, Oceanography, Parasitology, and Remote sensing'. The search yielded approximately 74,000 results. After a first title and abstract screening, we selected 972 relevant articles (see Supplementary Fig. 2 for PRISMA flowchart[32]). From these studies, we selected those that (i) were actually related to ecosystem recovery, (ii) had at least three measurements of recovery in time, (iii) had a clear reference system (either in the pre-disturbance state or an 'undisturbed' ecosystem with similar environmental conditions), (iv) were related to any of the nine disturbance categories, (v) reported time since recovery started and (vi) included measures of organism abundance, species diversity and cycling of carbon and nitrogen. We only considered these five recovery metrics to reduce the inherent heterogeneity of the database and provide robust results, particularly in the case of biogeochemical functions.

Our selection yielded 278 primary studies. From these studies, we extracted 3,468 comparisons of measurements of recovery between reference and recovering ecosystems from tables, figures and text of the paper. Outcome measures extracted from the selected studies were already averaged across several sampling plots in most cases. We used the free software DataThief III[33] to extract data from the figures. Following the same selection criteria, we added 253 outcome measures from the database of Rey Benayas et al.[8] and 95 from the database of Meli et al.[31]

totaling 3,816 outcome measures. We included each outcome measure separately, instead of averaging them per study, because we assumed independent responses of each parameter to the recovery process.

To ensure the quality of the data in the new data set, a protocol for data extraction was created and each person who entered data was trained with three manuscripts, to ensure accurate numbers were entered and accurate categorizations were made using the same form. H.P.J. met with data enterers bi-weekly throughout the data collection process to answer questions about conflictive data, data entry selection and spot-checked data entered throughout the process to ensure accuracy. At these meetings, data enterers had the opportunity to raise ambiguities or other issues found during the extraction process and any disagreements were resolved by consensus. Lastly, H.C.J., P.C.J. and D.M.M. checked each category assigned per study before the data were analysed, including the data sets from Rey Benayas et al.[8] and Meli et al.[31]

Studies used field-based measurements to assess ecosystem recovery of various outcome measures after disturbances. The outcome measures related to organism abundance included measurements of density, biomass, cover and basal area of trees, shrubs, grasses and algae, and measurements of density of birds, fish and invertebrates. The outcome measures related to diversity included mainly measurements of species richness and diversity indexes such as Shannon, Simpson and evenness indexes. Biogeochemical outcome measures related to the cycling of carbon and nitrogen contain both pools and fluxes of these elements in soil, litter and the water column. To test for potential differences between different kinds of measures within our metrics, we have estimated average effects sizes for subcategories within the metrics 'diversity', 'carbon cycling' and 'nitrogen cycling'. In the metric diversity, we defined subcategories 'species richness' and 'diversity indexes', this last one including Simpson, Shannon and evenness indexes. In the metrics carbon cycling and nitrogen cycling, we compared subcategories 'pools' and 'fluxes'. The subcategory pools ($n = 414$ for carbon and $n = 212$ for nitrogen) mostly included concentration of carbon or organic matter in soils or litter measured in weight units per volume units or weight units per area units. Fluxes ($n = 53$ for carbon and $n = 38$ for nitrogen) measured respiration, mineralization, accumulation, immobilization or decomposition rates in weight units of carbon or nitrogen per weight unit of soil or litter and time.

Species richness and diversity indexes had a marginal difference in their confidence intervals, richness 22.6–28.6% and diversity 28.8–34.6%. Even smaller differences were found between the pool and stock subcategories of carbon, stock 32.2–42.1% and pulse 37.4–47.4%, and nitrogen, stock 32.1–41.6% and pulse 38.9–48.8%. The largely skewed sample sizes between all subcategories did not allow to perform reliable Wald's tests. Although Mann–Whitney tests are not best adapted to test for significant differences in meta-analytic data, we found nonsignificant ($P > 0.1$) differences in any the subcategory tests performed. These marginal differences in the average effect sizes of the selected subcategories suggested that no major differences should be expected in the behaviour of each subcategory within each metric. Splitting these metrics into subcategories involved having substantially less robustness in the main analysis that prevented having reliable comparisons in most of the categories within the moderators ecosystem type and degrading factor. Thus, metrics were maintained undivided.

For each outcome measure, we also collected data on the climatic region according to the Köppen–Geiger climate classification system[34], number of sites undergoing recovery, number of reference sites, area of the study site, ecosystem category (forest, grassland, wetlands, river, lake or marine), disturbance category (one of the nine factors used in the search or multiple when more than one category was reported), disturbance duration and time since recovery started. Except in studies monitoring land cover change ($n = 3$), the size of these plots ranged from $< 1 \text{ m}^2$ to a few hectares. Even though the area of the study site was only reported in a limited number of studies (Supplementary Table 1), we collected these data to approximate the spatial representation of our results.

Regarding ecosystem category, forests included all ecosystems where trees were dominant, wetlands included both freshwater and coastal aquatic ecosystems according to the Ramsar Convention definition[35], grasslands included ecosystems where grasses and forbs were dominant and marine ecosystems included benthic and pelagic ecosystems from the shoreline to $> 100 \text{ m}$ deep. The number of sites undergoing recovery included the number of plots being measured but not replicates within plots. In the case of chronosequences, we only recorded data for the start and end time points. The total area resulted from adding the areas of all the study sites that were reported and thus our estimated total accumulated study area is a conservative estimate. As previous studies have reported that restoration approach (that is, passive versus active restoration) does not generate significantly different responses in wetlands over the long term[36], nor in other ecosystem types[22], we have excluded this factor. The disturbance duration was reported in 217 studies and ranged from $< 1$ day to 379 years (mean ± s.e., 29 ± 3.1 years; median = 6). The time since restoration started ranged from $< 1$ day to 380 years (mean ± s.e., 13 ± 1.9 years; median = 9).

**Weighting.** As is commonplace with ecological meta-analyses[8,10,37,38], the data necessary to determine variance with any confidence were not available in the majority (79% in our case) of outcome measures. In addition, meta-analysis theory suggests that when among-study variation is much higher than within-study variation, parameter estimates from random-effects models are nearly the same as

those obtained with unweighted models[37–40]. Nevertheless, unweighted models may yield confidence intervals that are too narrow, as they do not account for the within- and among-study variation components that are accounted for in random-effects models. In our meta-analysis, we used the subset of studies reporting variances (or enough information to calculate them) and computed the $I^2$ index[41] separately for each outcome variable. The values of $I^2$ were over 90% (range 93.74–99.64%), suggesting that among-study heterogeneity accounted for most variation and that random-effects weights across effect sizes would be expected to be very similar. Then, we used this subset of the database to estimate the average within-study variances as an arithmetic mean of the available within-study variances for each outcome variable and used the obtained values as approximate within-study variances for the remaining effect sizes, which had that information missing. This strategy allowed us to fit random-effects meta-analytic models for each outcome category.

**Quantification of the recovery debt.** Analytically, the recovery debt for a recovery process that takes place over a period of time $T$ (usually denoted in years) can be calculated using $X_s$ (the value of the relevant ecosystem metric at the start of the recovery phase, at time $t = 0$), $X_e$ (the value of the same metric at the end of the recovery period after a finite period of time, $t = T$) and $X_r$ (the reference value of the same metric in a reference system, either in the pre-disturbance state or in an 'undisturbed' equivalent system; Supplementary Fig. 5). The transition between $X_s$ and $X_e$ is unknown but likely to be nonlinear[42], which was approximated using an exponential function, $f(x) = e^{rt}$, where $r$ is constant and $t$ is the time between $X_s$ and any point between $X_s$ and $X_e$. We estimated the recovery debt for each of the selected outcome measures using both an exponential and a linear approximation and found small differences between the resulting recovery debt estimations (Supplementary Fig. 6), which led to similar conclusions.

There are five scenarios in which the recovery debt can be calculated (Supplementary Fig. 3). The value of the recovery debt is the area existing between the $X_r$ (reference value of the outcome measure) line and the line connecting $X_s$ (starting value) and $X_e$ (end value). According to Supplementary Fig. 5, in scenarios $a$ ($n = 1,993$) and $c$ ($n = 446$), the recovery debt area during the time period $(0, T)$ is $X_r T$–AUC (area under the curve, green shading). There were many cases where $X_s$ and $X_e$ were higher than $X_r$ or where $X_s > X_r > X_e$, represented in scenarios $b$ ($n = 424$) and $d$ ($n = 953$). Starting values exceeding reference values is commonly found for abundance measurements in early recovery stages and also for nitrogen concentration in aquatic ecosystems undergoing eutrophication. In these cases, we assumed that response values above the reference value represent negative effects, and thus $X_s$ and $X_e$ were inverse-transformed using the formula $Z_{s,e} = X_r \frac{X_r}{X_{s,e}}$. In the particular case where $X_r = 0$, scenario $e$ ($n = 236$), we used the approximation $Z_{s,e} = \frac{X_r^2}{X_{s,e}}$. This allowed us to compare those cases with the rest of the database and to set a realistic recovery threshold of 100%.

The AUC was calculated as $X_s\, e^{rT} = X_e$, it follows that $\ln e^{rT} = rT = \ln \frac{X_e}{X_s}$ and therefore $r = \frac{1}{T}\ln(\frac{X_e}{X_s})$. The AUC is then AUC $= \int_0^T e^{rt}\,dt$, where $r$ is defined above. After integration, AUC $= \frac{1}{r}X_s e^{rT} - \frac{1}{r}X_s = \frac{1}{r}[X_e - X_s]$.

It follows that recovery debt (RD) that occurs over the time period $(0, T)$ is:

$$\text{RD} = X_r T - \text{AUC} = X_r T - \frac{1}{r}[X_e - X_s]$$

It is noteworthy that as $T$ varies for each outcome measure, it is preferable to express recovery debt in per annum terms. That is, recovery debt per annum is:

$$\text{RDt} = \frac{\text{RD}}{T} = X_r - \frac{1}{rT}[X_e - X_s]$$

In scenarios $c$ and $d$, a negative exponential function $f(x) = e^{-rt}$ is assumed and used to estimate the recovery debt. In scenario $e$, RDt $= \frac{\text{AUC}}{T}$.

Given the heterogeneity of the outcome measure measured, we homogenized the values of RDt to compare them. We did so by estimating the recovery debt ratio as a percentage RDr(%) $= 100 * \frac{\text{RDt}}{\text{Abs}(X_r)}$, where Abs($X_r$) is the absolute value of $X_r$. In a some cases ($n = 250$), the value of RDt $> X_r$ and, following the same principle used to estimate $Z_{s,e}$, the recovery debt value was estimated as RDr(%) $= 100 * \frac{\text{Abs}(X_r)}{\text{RDt}}$.

The same approach was used to estimate the recovery debt value with the linear approximation. To estimate the area representing the recovery debt value, we used a linear function that merges any two points with positive slope. After calculation, the resulting formula is RDt $= X_r - 0.5*(X_e + X_s)$. We also used a linear function that merges any two points with negative slope and after calculation the resulting formula is RDt $= X_r - 1.5*X_e + 0.5*X_s$. Finally, when $X_r = 0$, we approximated the recovery debt ratio as RDr(%) $= 100*(1 - \text{RDt})$.

The presence of zero values in the outcome measures could produce abnormally high or low values of our recovery debt estimations[43]. This issue occurred in two situations: (i) when $X_r = 0$ (scenario $e$) and (ii) when $X_s$ or $X_e = 0$ ($n = 628$). To select the best approach to tackle this issue, we tested nine different strategies (Supplementary Table 3) to calculate the $r$ parameter required to estimate the recovery debt. From them, six included the use of a constant value added to the numerator and denominator (that is, 0.01, 0.05, 0.1, 0.5 and 1) of the formula used to calculate $r$ and four included the use of an amount specific for each outcome measure. These specific amounts were (i) an amount of the same order of magnitude than $X_s$ and $X_e$ located at the beginning of the order of magnitude (for example, 0.1, 1 and 10), (ii) the same than (i) but with the median magnitude (for

example, 0.5, 5 and 50), (iii) an amount one order of magnitude larger than $X_s$ and $X_e$ located at the beginning of the order of magnitude and (iv) the same as (iii) but with the median magnitude. For example, if $X_s = 0.81$, the four amounts were 0.1, 0.5, 1 and 5, respectively. We compared the data distribution (median and 95% confidence interval) of each strategy with the rest of the database without the zero values using Mann–Whitney rank sum tests. Only the approach using an amount of the same order of magnitude than $X_s$ and $X_e$ at the median magnitude had a nonsignificant data distribution ($P > 0.05$) different from the rest of the database and thus this was the strategy we used in the two situations where we needed to address zero values. This strategy was not used in the rest of the database not affected by zero values. To further ensure that there were no differences between this approach and an approach that simply excludes all zero values, we compared the recovery debt excluding and including outcome measures with zero values and we did not find qualitative differences that could lead to different conclusions (Supplementary Fig. 7).

**Statistical approach.** We ran sensitivity analyses using generalized linear models with a variety of probability distributions (that is, normal, log-normal and Gamma), link functions (that is, identity and log) and rescaling the data to determine the best modelling approach for this analysis. Through all of the models, the general mixed model with normal distribution and unscaled data was the most parsimonious and the best fit to the data in terms of Akaike information criteria (AIC) and likelihood ratio tests, and thus we continued to use a meta-analytic mixed model approach based on the normal distribution. We used the multivariate, mixed model function $rma.mv()$ from the *metafor* package[44] to construct three-level meta-analytic models in R 3.0.1 (ref. 45). We used three-level models, because our units of analysis (recovery debt) were clustered within effect sizes and those effect sizes were clustered within studies.

Following Mergensen et al.[37], 'homogeneity tests are usually not undertaken and are not meaningful, in cases where a random-effects model has been used for conceptual reasons and/or because the meta-analyst recognizes in advance that there is substantial between-study variation'. Therefore, we did not carry out a heterogeneity test, although we estimated the $I^2$ index for each outcome category and the resulting values show evidence of substantial heterogeneity between study results, which is accounted in our models as explained in the 'Weighting' section. We divided the data into four subsets based on the type of metric that was measured: abundance, diversity, carbon cycling and nitrogen cycling. We investigated the effect of moderators on our models with a three-step approach[44]. We first fit a three-level meta-analytic model without moderators. Then, we tested the significance of the moderators using the omnibus test of moderators ($Q_M$ test). Third, we fit a model without intercept to get the effect size estimates and confidence intervals for each category of the moderator variable. We estimated the overall effect sizes and confidence intervals of recovery debt in each subset without any moderating effects from the null models.

**Data availability.** Data including the database used and the codes generated in R are provided in the Dryad Digital Repository, doi:10.5061/dryad.t5c97.

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

## Acknowledgements

We acknowledge Jessica Gurevitch and Karen Holl for comments, and Carlos Ruiz for the map. This work was supported by the National Socio-Environmental Synthesis Center (SESYNC) under funding received from the National Science Foundation DBI-1052875, by the German Helmholtz Centre for Environmental Research–UFZ, Leipzig (Research Program 'Terrestrial Environments') and by sDiv, the Synthesis Centre of the German Centre for Integrative Biodiversity Research (iDiv) Halle-Jena-Leipzig (German Research Foundation DFG FZT 118).

## Author contributions

D.M.-M. and E.B.B. designed the study and wrote the manuscript. P.C.J., J.A.L.-L. and D.M.-M. analysed the data. H.P.J. and D.M.-M. coordinated the construction of the database. H.P.J., D.M.-M., J.M.R.B., P.C.J., M.L.M., P.M.and D.M. collected the data and edited the manuscript. P.C.J., J.A. and J.A.L.-L. edited the manuscript.

## Additional information

**Competing financial interests:** The authors declare no competing financial interests.

**Publisher's note**: 

