## [Peer Review File · Nature Communications]

Reviewers' comments:

Reviewer #1 (Remarks to the Author):

The question and the approach appear to be appropriate and the results are useful. The problem is that the wording of the manuscript is confusing in many places, and I was unsure what was being compared and concluded. The claim that there is no systematic review seems overstated. It's not a necessary rationale for the study; a larger meta-analysis than ones that have been done before still has utility.

Examples of unclear terms follow: "response variable" is used in two ways; we are told there are 5 response variables (when the list gives 4; I suspect that was a typo); then we are told there are over 3,000 response variables (word misuse). Throughout the ms, diversity, biodiversity, and species richness are used without clarification, although we are told that diversity refers to indices that include abundance. "Abundance" needs to be clearer in the text, especially for organisms that include algae. Significance = non-overlapping C.L. on page 2, but non-overlapping s.e. at the end of the methods. ?

I put comment notes on the text to suggest substitute words where I could guess the intent. I highlighted in yellow the words and phrases that were not clear.

I suggest that the authors provide some suggestion at the end indicating how to deal with the inevitable recovery debt. To fulfill wetland "no-net-loss" policy in USA, this debt has been dealt with by increasing the mitigation ratio so that more area of less-functional wetlands supposedly compensates for the debt. I thought the authors might wish to comment on the relevance of their quantification of debt to the practice of increasing the ratio to compensate for well-known shortcomings of restored wetlands. Best regards, Joy Zedler

Reviewer #2 (Remarks to the Author):

This ms. reports on a very large systematic review and meta-analysis on the 'recovery debt'-the reduction in biodiversity and ecosystem processes following disturbance and during recovery. The authors consider 9 types of disturbances, six major types of ecosystems, four broad categories of response, and five categories of organisms. The problem addressed is of high general interest and importance, and the paper clearly represents an enormous amount of time and effort. Unfortunately I am not convinced by the detailed results (although I suspect that the general conclusions are generally correct), for the following reasons. First, the meta-analysis is unweighted, although sample sizes appear to range from 9 to 130 (Extended Data Table 1) if in fact these are the sample sizes used (it was a little bit difficult to tell). That means that the studies varied enormously in the precision with which recovery debt was estimated, and which is why meta-analyses are (and should be) weighted by the inverse of their precision. Second, the effect sizes include such a broad range of types of measures of response that I am not confident that they are meaningful; for example, diversity indices are not reasonable to combine with species richness measures. The paper even hints that they tell you very different things (p. 2 lines 33-35). It is not clear how different other measures of response were that were combined into single average effects, but it seems likely that this was also done for other measures of outcome. What one decides to combine into a single average measure of response in a meta-analysis has to be very carefully thought through and well justified, and I was not convinced by the limited justification presented that this really was reasonable and gave informative results. Combining recovery debt across such different types of organisms is even more problematic-I can't even imagine what it means to report a single average response of invertebrates, algae, trees and birds to mining, for instance, and I am sure that this average response includes very different types and numbers of organisms for the different habitats and disturbances, so that makes its meaning even more problematic. Third, no tests of heterogeneity were carried out, and assuming that heterogeneity

was very large, it is even less clear that the means presented are truly meaningful. Yet, the responses are presented as if they are highly accurate and meaningful (e.g. the recovery debt for species diversity was 31% +/- 1.5%); the confidence interval does not provide an accurate estimate of the heterogeneity among studies. Even beyond heterogeneity statistics, though, what is going into the blender here is unlikely to give truly interpretable results, and certainly not results that are accurate and meaningful to a decimal place of a percent. Fourth, I did not see any theoretical derivation of the probability density function of the effect size metric used here, or whether it is likely to be normally distributed or far from normally distributed (although it is somewhat related to response ratios, which suggests that it might be log normally distributed), so I was not confident about the results of the meta-regressions, even without considering the other issues discussed above. Finally, the statistical analyses should provide tests of which values of the moderators were actually different; model selection might also be used here. So, the statements based on the confidence intervals (e.g. "Mined sites had a higher diversity debt than sites affected by hydrological disruptions and hurricanes") were unconvincing. I would really like to be convinced by the results of this study, because, again, the questions being raised are really important and of high interest and there was a huge amount of work involved here, but the analyses did not give me confidence in the results.

More minor comments are:

I did not understand what this meant: "We only included data in the models when at least 15 values of recovery debt from three studies for each level in the moderator variables ..."

I did not fully understand what was meant by "we extracted 3,468 response variables ". Does this mean (as I think it means) 3,468 different results included in the meta-analysis? It sounds like there were thousands of different ways that the responses were measured, which would be unfortunate to say the least, but I don't think that's really what is meant here.

It would be very useful to provide the full outcome of the systematic review in the format used by PRISMA as required in other disciplines (although not yet for ecological research syntheses).

Will the complete data set be made available in a reliable repository once the paper is published? That would be essential.

REVIEWERS' COMMENTS:

Reviewer #2 (Remarks to the Author):

This revision is substantially improved over the previous ms., and the authors put considerable effort into addressing the comments. I still have two fundamental reservations about the paper, one conceptual and one statistical. The first is whether it makes sense to combine so many different measures of outcome on so many different organisms and systems. While I am very strongly supportive of the power to generalize results using meta-analysis, and I understand that some comparisons between groups and metrics were not statistically significantly different, that still doesn't resolve the basic issue of what these combined results mean. The way the results are worded is far better than the previous version, presenting a range of outcomes and reporting I^2 values, and to some extent this is a matter of personal philosophy and judgement, rather than being scientifically faulty or 'wrong'. Are such heterogeneous results on such heterogeneous systems, measures of outcome and organisms meaningful? I don't know. I think the way this is now reported is transparent and largely clear, so readers can reach their own evaluation of the outcomes. Perhaps even more caution in interpreting the results in light of the vast range of things being combined and the heterogeneity of the outcomes would be valuable.

Statistically, I am not clear on whether the non-independence among outcomes (plots within studies, among organisms, etc.) were corrected for or how they were taken into account in the construction of the confidence intervals. If 'plots' are the unit, is the non-independence for plots within studies taken into account/corrected in the estimation of confidence intervals? It seems very strange actually that plots would be used as the units in the analyses. Are the confidence intervals really of the stated magnitude?

Other comments are as follows:

"Response variable" is really kind of ambiguous as a term. Would "outcome measures" be a better way to express what you mean here? "Response variable" implies different metrics of response, not the measured outcomes of response from each plot (e.g. two response variables that might be measured on a single plot could be biomass and soil nitrogen). I don't think there were thousands of different kinds of outcomes measured; rather, I think some of the same response variables were measured on many plots-- for instance, species richness (a response variable) was clearly measured on many different plots, giving many different outcome measures from different plots for the variable species richness, rather than many response variables.

Inter-rater agreement is certainly not just for subjective data sets (line 180); see J. Littell, Systematic Reviews and Meta-analysis for an extended discussion on inter-rater practices.

Disturbance duration and time since disturbance should be reported as medians as well as means (lines 229-231); it seems likely that these are non-normally distributed and in any case medians would be of interest in addition to the means and CIs.

Lines 244-245: The paper states, "Then, we used this subset of the database to estimate the average within-study variances, and used the obtained values to approximate within-study variances for the remaining effect sizes." How was that done? It was not clear to me what was done. Please clarify.

Line 269: Do you mean X_r ? I don't see X_g defined. Various other typos (e.g. line 95, 'cold' for 'could') occur in the text.

Lines 260-270: Doesn't this bias the results?

I don't think testing the data fit addresses the issue of the sampling distribution of the effect size being used. The latter is derived from probability theory, not from tests on data.

Line 331: this is most certainly not a "null model" in any sense of the term (statistical or ecological). Please change the wording.

Line 46-47: I think the question is not "if" but rather, "to what extent" (it is not a yes or no question).

Response to reviewers' comments (highlighted in red)

Reviewer #1 (Remarks to the Author):

The question and the approach appear to be appropriate and the results are useful. The problem is that the wording of the manuscript is confusing in many places, and I was unsure what was being compared and concluded. The claim that there is no systematic review seems overstated. It's not a necessary rationale for the study; a larger meta-anal than ones that have been done before still has utility.

We agree with the reviewer, and have rewritten sentences in both the Abstract and the manuscript as follows:

Abstract: As ecosystems progress through recovery, it is important to estimate any resulting deficit in biodiversity and functions.

Main text: Assessments of anthropogenic disturbances have shown global losses¹¹ in biodiversity while the disturbance is still active and time lags in its response^{12,13} (Fig. 1). However, as ecosystems recover after the disturbance ceases, it is less clear whether they continue to endure deficits in biodiversity and functionality.

Examples of unclear terms follow: "response variable" is used in two ways; we are told there are 5 response variables (when the list gives 4; I suspect that was a typo); then we are told there are over 3,000 response variables (word misuse).

This inconsistency has been corrected throughout the manuscript. Recovery metric now refer strictly to the five variables for which we have data, those are, abundance, diversity, and carbon and nitrogen cycling. Response variable refers to the raw data we used to estimate the recovery debt, that is, each data-point entered from the selected studies.

Throughout the ms, diversity, biodiversity, and species richness are used without clarification, although we are told that diversity refers to indices that include abundance. "Abundance" needs to be clearer in the text, especially for organisms that include algae.

In addition to the existing description of the response metrics in the methods, we have now added a short description in the main text p. 2, l. 29-35.

Significance = non-overlapping C.L. on page 2, but non-overlapping s.e. at the end of the methods. ?

We are no longer using this method to detect significant differences. Both sentences have been completely changed according to the new methodology.

I put comment notes on the text to suggest substitute words where I could guess the intent. I highlighted in yellow the words and phrases that were not clear.

All the comment notes have been corrected.

I suggest that the authors provide some suggestion at the end indicating how to deal with the inevitable recovery debt. To fulfill wetland "no-net-loss" policy in USA, this debt has been dealt with by increasing the mitigation ratio so that more area of less-functional wetlands supposedly compensates for the debt. I thought the authors might wish to comment on the relevance of their quantification of debt to the practice of increasing the ratio to compensate for well-known shortcomings of restored wetlands. Best regards, Joy Zedler.

We agree with this suggestion, and have rewritten the last few sentences of the main text in the following way, and added a new reference (29. J. Zedler, S. Kercher. Wetland Resources: Status, Trends, Ecosystem Services, and Restorability. *Annu. Rev. Environ. Resour.* **30**, 39-74 (2005):

“Under such circumstances, ecological restoration and offsetting are inadequate alternatives to ecosystem protection. This would also suggest reconsidering restoration policies that attempt to fulfill “no-net-loss” principles by simply increasing the mitigating or offset ratio so that more area of less-functional ecosystems are created²⁹. If the restoration debt is large and sustained over several decades, then increasing the quantity of less-functional ecosystems is poor compensation for the overall intervening loss in ecosystem biodiversity and functions.”

Reviewer #2 (Remarks to the Author):

This ms. reports on a very large systematic review and meta-analysis on the 'recovery debt'-the reduction in biodiversity and ecosystem processes following disturbance and during recovery. The authors consider 9 types of disturbances, six major types of ecosystems, four broad categories of response, and five categories of organisms. The problem addressed is of high general interest and importance, and the paper clearly represents an enormous amount of time and effort. Unfortunately I am not convinced by the detailed results (although I suspect that the general conclusions are generally correct), for the following reasons.

First, the meta-analysis is unweighted, although sample sizes appear to range from 9 to 130 (Extended Data Table 1) if in fact these are the sample sizes used (it was a little bit difficult to tell). That means that the studies varied enormously in the precision with which recovery debt was estimated, and which is why meta-analyses are (and should be) weighted by the inverse of their precision.

We agree with the reviewer that weighting is a crucial issue in meta-analysis. Despite the limited information available in our database, we have made additional efforts and have added a whole new subsection in the Methods section. It now reads as follows:

“Weighting. As is commonplace with ecological meta-analyses (Rey Benayas et al. 2009, Moreno-Mateos et al. 2012, Mergensen et al. 2013, Crouzeilles et al. 2016), the data necessary to determine variance with any confidence were not available in the majority (79% in our case) of response variables. Also, meta-analysis theory suggests that when among-study variation is much higher than within-study variation, parameter estimates from random-effects models are nearly the same as those obtained with unweighted models (Hedges et al. 1999, Gurevitch et al. 2001, Mergensen et al. 2013). Nevertheless, unweighted models may yield confidence intervals that are too narrow, as they do not account for the within- and among-study variation components that are accounted for in random-effects models.

In our meta-analysis, we used the subset of studies reporting variances (or enough information to calculate them) and computed the I^2 index (Higgins and Thompson 2002) separately for each outcome variable. The values of I^2 were over 90% (range 93.74% to 99.64%), suggesting that among-study heterogeneity accounted for most variation and that random-effects weights across effect sizes would be expected to be very similar. Then, we used this subset of the database to estimate the average within-study variances, and used the obtained values to approximate within-study variances for the remaining effect sizes. This strategy allowed us to fit random-effects meta-analytic models for each outcome category.”

In any event, it seems we were unclear in explaining the sample sizes we used to estimate the recovery debt values in Supplementary Table 1. The sample sizes the reviewer refers to are the number of studies

we had in the database for each of the categories, not the ones we used to estimate the recovery debt values. Those sample sizes are the number of response variables used to calculate average effect size and confidence interval estimates and are included in each of the bars in Figure 2 of the manuscript. To clarify this, we have added two columns to Supplementary Table 1 with data on the distribution of the response variables across our categories.

Second, the effect sizes include such a broad range of types of measures of response that I am not confident that they are meaningful; for example, diversity indices are not reasonable to combine with species richness measures. The paper even hints that they tell you very different things (p. 2 lines 33-35). It is not clear how different other measures of response were that were combined into single average effects, but it seems likely that this was also done for other measures of outcome. What one decides to combine into a single average measure of response in a meta-analysis has to be very carefully thought through and well justified, and I was not convinced by the limited justification presented that this really was reasonable and gave informative results. Combining recovery debt across such different types of organisms is even more problematic-I can't even imagine what it means to report a single average response of invertebrates, algae, trees and birds to mining, for instance, and I am sure that this average response includes very different types and numbers of organisms for the different habitats and disturbances, so that makes its meaning even more problematic.

We understand the concerns of the reviewer about merging different metrics and agree that differences in the response can be expected. To test for potential differences between different kinds of measures within our categories, we have estimated average effects sizes for subcategories within the categories “diversity”, “carbon cycling”, and “nitrogen cycling”. In the category diversity, we defined subcategories “species richness” and “diversity indexes”, this last one including Simpson, Shannon and evenness indexes. In the categories carbon cycling and nitrogen cycling, we compared subcategories “pools” and “fluxes”. The subcategory pools mostly included concentration of carbon or organic matter in soils or litter measured in weight units per volume units or weight units per area units. Fluxes measured respiration, mineralization, accumulation, immobilization or decomposition rates in weight units of carbon or nitrogen per weight unit of soil or litter and time. Species richness and diversity indexes had a marginal difference in their confidence intervals, richness 22.6 – 28.6 and diversity 28.8 – 34.6. Even lower differences were found between the pool and stock subcategories of carbon, stock 32.2 – 42.1 and pulse 37.4 – 47.4, and nitrogen, stock 32.1 – 41.6 and pulse 38.9 – 48.8. The largely skewed sample sizes between all subcategories did not allow to perform reliable Wald tests. Although Mann-Whitney tests are not best adapted to test for significant differences in meta-analytic data, we found non-significant ($p > 0.1$) differences between all the subcategory tests performed.

These marginal differences suggested that no major differences should be expected in the behavior of different subcategories within each metric. Splitting these metrics into subcategories involved having substantially less robustness in the main analysis that prevented having reliable comparisons in most of the categories within the moderators ecosystem type and degrading factor. Where to stop splitting is a common concern in meta-analysis, especially when associated to large databases with heterogeneous data. In this particular case, the loss of information was so high and the benefits so uncertain, given the marginal difference between subcategories, that we assumed certain loss of accuracy for the benefit of robustness. We have however included this analysis and clarified our metric selection in the methods section (p. 5, l. 11-43).

We have included a new analysis to see the effect of a category named “organism type” that includes tree/shrub, invertebrate, grass/herb, bird/fish, and algae and looked for differences between them across ecosystem types and disturbance categories (Supplementary Figs. 5 and 6). Again, we found no evidence

of moderating effects of the “organism type” moderator, which justifies not splitting them by organism category. We have also added a sentence in the main text clarifying this (p. 3, l. 9-10).

Even after these results, we acknowledge that our database has different measures. For example, abundance was measured in different ways including, biomass, number of individuals/stems, or percent cover. But in all cases, the question was quite similar, how much did the population size change after restoration? Similarly, we had three other questions, how much diversity has changed? And how much carbon and nitrogen is now cycled? By using our recovery debt metric, the differences between measures were standardized and made comparable in a similar way to what response ratios do. Because of this, we think we can say we have more or less organisms or diversity in restored ecosystem as compared to reference ones and show it for different organism types, ecosystem types, and degradation categories. We cannot be specific, and we do not want to be, with this dataset, but the results are consistent and at least we can show some general trends that are meaningful and hopefully helpful.

Third, no tests of heterogeneity were carried out, and assuming that heterogeneity was very large, it is even less clear that the means presented are truly meaningful. Yet, the responses are presented as if they are highly accurate and meaningful (e.g. the recovery debt for species diversity was 31% +/- 1.5%); the confidence interval does not provide an accurate estimate of the heterogeneity among studies. Even beyond heterogeneity statistics, though, what is going into the blender here is unlikely to give truly interpretable results, and certainly not results that are accurate and meaningful to a decimal place of a percent.

We thank the reviewer for bringing up this important issue. As mentioned above, we reran our analyses taking heterogeneity into consideration, and made the most of the limited information available in our meta-analytic database. Following Mergensen et al. (2013): “homogeneity tests are usually not undertaken, and are not meaningful, in cases where a random-effects model has been used for conceptual reasons and/or because the meta-analyst recognizes in advance that there is substantial between-study variation”. Therefore, we did not carry out a heterogeneity test, although we estimated the I^2 index for each outcome category (see response to question 1) and the resulting values show evidence of substantial heterogeneity between study results. Our updated meta-analytic models account for such heterogeneity, and this is reflected in wider confidence intervals. Moreover, we now acknowledge in the discussion that the presence of substantial between-study heterogeneity requires a cautious interpretation of our results (p. 3, l. 36-38). Also, we have now included confidence intervals through the text to address the decimal concerns of the reviewer.

Fourth, I did not see any theoretical derivation of the probability density function of the effect size metric used here, or whether it is likely to be normally distributed or far from normally distributed (although it is somewhat related to response ratios, which suggests that it might be log normally distributed), so I was not confident about the results of the meta-regressions, even without considering the other issues discussed above.

We apologize for not being stricter with the normality test. This comment has been crucial to improve the robustness of the statistical analysis and thus of the whole manuscript. We ran sensitivity analyses using generalized linear models with a variety of probability distributions (i.e. normal, log-normal, Gamma), link functions (i.e. identity and log), and rescaling the data to determine the best modeling approach for this analysis. Through all of these different models, the general mixed model with normal distribution and unscaled data was the most parsimonious and the best fit to the data in terms of AIC and likelihood ratio tests, and thus we continued to use a meta-analytic mixed model approach based on the normal distribution. This approach is now part of the methods section (p. 8, l. 23-38).

Finally, the statistical analyses should provide tests of which values of the moderators were actually different; model selection might also be used here. So, the statements based on the confidence intervals (e.g. "Mined sites had a higher diversity debt than sites affected by hydrological disruptions and hurricanes") were unconvincing.

Again, we thank the reviewer for bringing this issue up and agree that relying solely on CI to make significance statements may present problems in certain cases (Schenker and Gentleman 2001, Cumming and Finch 2005). To understand the effect of moderators on our models, we used the three step approach presented in Viechtbauer (2009). We first fit a 3-level meta-analytic model without moderators (null model); then, we tested the significance of the moderators using the QM test; and third, we fit a model without intercept to get the effect size estimate (and CI) for each category. This approach is now included in the "Statistical approach" section within the Methods (p. 8, l. 33-46)

I would really like to be convinced by the results of this study, because, again, the questions being raised are really important and of high interest and there was a huge amount of work involved here, but the analyses did not give me confidence in the results.

More minor comments are:

I did not understand what this meant: "We only included data in the models when at least 15 values of recovery debt from three studies for each level in the moderator variables ..."

Corrected. It now reads: "We used model estimations to calculate average recovery debts when we had at least 15 response variables from at least three studies".

I did not fully understand what was meant by "we extracted 3,468 response variables ". Does this mean (as I think it means) 3,468 different results included in the meta-analysis? It sounds like there were thousands of different ways that the responses were measured, which would be unfortunate to say the least, but I don't think that's really what is meant here.

Corrected. It now reads: "we extracted 3,468 comparisons of measurements of recovery between reference and recovering ecosystems".

It would be very useful to provide the full outcome of the systematic review in the format used by PRISMA as required in other disciplines (although not yet for ecological research syntheses).

Since the database used for this meta-analysis includes response variables coming from three different databases (one newly created for this study and two previously published (Rey Benayas et al. 2009, Meli et al. 2014)), we have added Supplementary Figure 2 with the PRISMA flowchart only for the studies that are new to this study. We did not include the studies from the previously published meta-analyses.

Will the complete data set be made available in a reliable repository once the paper is published? That would be essential.

Yes, the complete dataset will be made available at the Dryad website once the manuscript is accepted.

References

Crouzeilles, R., M. Curran, M. S. Ferreira, D. B. Lindenmayer, C. E. V Grelle, and J. M. Rey Benayas. 2016. A global meta-analysis on the ecological drivers of forest restoration success. *Nat Commun* 7:1–8.

Cumming, G., and S. Finch. 2005. Inference by eye: confidence intervals and how to read pictures of

- data. *The American psychologist* 60:170–180.
- Gurevitch, J., P. S. Curtis, and M. H. Jones. 2001. Meta-analysis in ecology. *Advances in ecological research* 32:199–247.
- Hedges, L. V., J. Gurevitch, and P. S. Curtis. 1999. The meta-analysis of response ratios in experimental ecology. *Ecology* 80:1150–1156.
- Jones, H. P., P. C. Jones, E. B. Barbier, R. C. Blackburn, J. M. R. Benayas, K. D. Holl, M. McCrackin, P. Meli, D. Montoya, and D. Moreno-Mateos. (n.d.). Restoration and repair of Earth's damaged ecosystems. *Nature Communications*.
- Meli, P., J. M. Rey Benayas, P. Balvanera, and M. Martínez Ramos. 2014. Restoration enhances wetland biodiversity and ecosystem service supply, but results are context-dependent: a meta-analysis. *PloS one* 9:e93507.
- Mergensen, K., C. H. Schmid, M. D. Jennions, and J. Gurevitch. 2013. Statistical Models and Approaches to Inference. Pages 89–107 *Handbook of Meta-analysis in Ecology and Evolution*.
- Moreno-Mateos, D., M. E. Power, F. A. Comín, and R. Yockteng. 2012. Structural and functional loss in restored wetland ecosystems. *PLoS Biology* 10:e1001247.
- Rey Benayas, J. M., A. C. Newton, A. Diaz, and J. M. Bullock. 2009. Enhancement of biodiversity and ecosystem services by ecological restoration: a meta-analysis. *Science* 325:1121–4.
- Schenker, N., and J. F. Gentleman. 2001. On Judging the Significance of Differences by Examining the Overlap Between Confidence Intervals. *The American Statistician* 55:182–186.

REVIEWERS' COMMENTS:

Reviewer #2 (Remarks to the Author):

This revision is substantially improved over the previous ms., and the authors put considerable effort into addressing the comments. I still have two fundamental reservations about the paper, one conceptual and one statistical.

The first is whether it makes sense to combine so many different measures of outcome on so many different organisms and systems. While I am very strongly supportive of the power to generalize results using meta-analysis, and I understand that some comparisons between groups and metrics were not statistically significantly different, that still doesn't resolve the basic issue of what these combined results mean. The way the results are worded is far better than the previous version, presenting a range of outcomes and reporting I^2 values, and to some extent this is a matter of personal philosophy and judgement, rather than being scientifically faulty or 'wrong'. Are such heterogeneous results on such heterogeneous systems, measures of outcome and organisms meaningful? I don't know. I think the way this is now reported is transparent and largely clear, so readers can reach their own evaluation of the outcomes. Perhaps even more caution in interpreting the results in light of the vast range of things being combined and the heterogeneity of the outcomes would be valuable.

We have modified the caveats paragraph to include this concern (l. 153-167 in the track changes version of the manuscript). In particular, we have emphasized the reviewer's concern that combining heterogeneous indicators into an aggregate indicator must be cautiously interpreted.

Statistically, I am not clear on whether the non-independence among outcomes (plots within studies, among organisms, etc.) were corrected for or how they were taken into account in the construction of the confidence intervals. If 'plots' are the unit, is the non-independence for plots within studies taken into account/corrected in the estimation of confidence intervals? It seems very strange actually that plots would be used as the units in the analyses. Are the confidence intervals really of the stated magnitude?

We are not using individual measurements from plots from each study, we are using their final response variables that should already take into consideration any non-independence among measurements among plots. We have clarified this now (l. 215-216). For further clarification, we used mixed-effects models to account for the hierarchical structure of our data set, so that the point estimates that we report are not expected to be affected by dependence issues. Regarding confidence intervals, as it is often the case in ecology, we did not have enough information to estimate within-study variances for all effect sizes. However, as we explain in the weighting section (which has now been expanded), the I^2 values (obtained using the subset of the database that included variance estimates) suggested that among-study heterogeneity accounted for most variation and that random-effects weights across effect sizes would be expected to be very similar. Due to this finding, we decided to use an approximation of the within-study variance for the studies that did not report it. This approximation was the average within-study variance for the effect sizes that included this information. Taking this approach enabled us to incorporate weights to our mixed-effects models, which resulted in wider confidence intervals that reflect not only the hierarchical structure of the data, but also the contribution of both within-study and among-study variation.

Other comments are as follows:

"Response variable" is really kind of ambiguous as a term. Would "outcome measures" be a better way to express what you mean here? "Response variable" implies different metrics of response, not the measured outcomes of response from each plot (e.g. two response variables that might be measured on a single plot could be biomass and soil nitrogen). I don't think there were thousands of different kinds of outcomes measured; rather, I think some of the same response variables were measured on many plots-- for instance, species richness (a response variable) was clearly measured on many different plots, giving many different outcome measures from different plots for the variable species richness, rather than many response variables.

We agree with the reviewer that “outcome measure” would be clearer and has been changed throughout the manuscript.

Inter-rater agreement is certainly not just for subjective data sets (line 180); see J. Littell, Systematic Reviews and Meta-analysis for an extended discussion on inter-rater practices.

We agree with the reviewer that our statement was wrong. We have rephrased the paragraph and removed the wrong statement (l. 222-230). We have explained in more detail our approach to deal with inter-rater agreement during the process of database construction.

Disturbance duration and time since disturbance should be reported as medians as well as means (lines 229-231); it seems likely that these are non-normally distributed and in any case medians would be of interest in addition to the means and CIs.

The median values have now been added.

Lines 243-245: The paper states, "Then, we used this subset of the database to estimate the average within-study variances, and used the obtained values to approximate within-study variances for the remaining effect sizes." How was that done? It was not clear to me what was done. Please clarify.

We have clarified this (l. 297-301).

Line 269: Do you mean Xr? I don't see Xg defined. Various other typos (e.g. line 95, 'cold' for 'could') occur in the text.

Corrected.

Lines 260-270: Doesn't this bias the results?

This is a new concern that should have been raised in the first review. We decided to do this because it was indeed not using this approach that was biasing the results making the debts higher and more variable. Given the ambiguity of this comment, we cannot think of any way this could bias the results.

I don't think testing the data fit addresses the issue of the sampling distribution of the effect size being used. The latter is derived from probability theory, not from tests on data.

Data fit via AIC is part of how you determine which probability distribution is the best one to use for your models. We are unsure why the reviewer says that it is not.

Line 331: this is most certainly not a “null model” in any sense of the term (statistical or ecological). Please change the wording.

The term has been removed (l. 388)

Line 46-47: I think the question is not “if” but rather, “to what extent” (it is not a yes or no question).

Changed (l. 47).